# Self-Supervised Multisensory Pretraining for Contact-Rich Robot Reinforcement Learning

*Abstract*— **Effective contact-rich manipulation requires robots to synergistically leverage vision, force, and proprioception. However, Reinforcement Learning agents struggle to learn in such multisensory settings, especially amidst sensory noise and dynamic changes. We propose MultiSensory Dynamic Pretraining (MSDP), a novel framework for learning expressive multisensory representations tailored for task-oriented policy learning. MSDP is based on masked autoencoding and trains a transformer-based encoder by reconstructing multisensory observations from only a subset of sensor embeddings, leading to cross-modal prediction and sensor fusion. For downstream policy learning, we introduce a novel asymmetric architecture, where a cross-attention mechanism allows the critic to extract dynamic, task-specific features from the frozen embeddings, while the actor receives a stable pooled representation to guide its actions. Our method demonstrates accelerated learning and robust performance under diverse perturbations, including sensor noise, and changes in object dynamics. Evaluations in multiple challenging, contact-rich robot manipulation tasks in simulation and the real world showcase the effectiveness of MSDP. Our approach exhibits strong robustness to perturbations and achieves high success rates on the real robot with as few as 6,000 online interactions, offering a simple yet powerful solution for complex multisensory robotic control. Video: https://drive.google.com/file/d/1yTEaamEGIEMbqjTLPo8Ad-2UF4534IRC/view**

## I. INTRODUCTION

Reinforcement Learning (RL) has shown impressive successes in learning complex tasks ranging from Atari [1], locomotion [2], vision-based manipulation [3] to multisensory peg insertion [4], [5]. However, incorporating multiple sensor modalities, especially in complex contact-rich robotic manipulation tasks, remains a challenge for RL, due to the heterogeneous dynamics of different sensor modalities. Additionally, the importance of each input modality changes during the execution of a manipulation task, e.g., coarse scene understanding from visual input to fine-grained force feedback when in contact. Thus, robotic agents need to learn how to dynamically focus on the most relevant sensory information while adapting to perturbations and dynamic changes in the environment. This field of *sensor fusion* is a long-studied problem in the field of robotics for various tasks ranging from control [6], [7], manipulation [8], [9], localization, and navigation [10], but remains underexplored in RL settings.

To this end, Imitation Learning approaches [11], [12], [13] have shown promise in utilizing multisensory data for learning skills, but require experts collecting informative data, thus limiting their usability, robustness and generalization. Self-directed exploration in RL, on the other hand, facilitates the

Accepted at IEEE Robotics and Automation Letters.

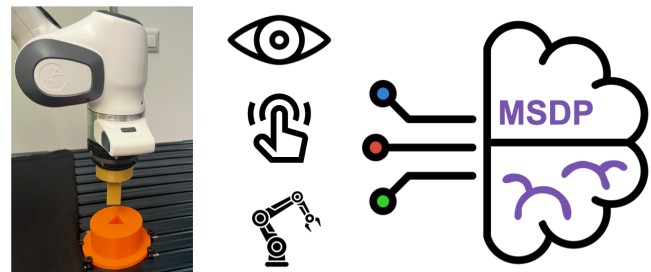

Fig. 1: Multisensory Dynamic Pretraining fuses multiple sensors, like human senses, to solve complex contact-rich manipulation tasks.

development of adaptable strategies that can be generalized across diverse object properties and contexts. These advantages position RL as a compelling approach for learning multisensory contact-rich manipulation [4], [14], [15]. Recent advances in multimodal learning [16], [17] have shown the advantages of reconstructing masked or noised inputs to learn expressive cross-modal representations for downstream tasks. Such masking-based self-supervision approaches, whether at the sensory or embedding level, have also been shown to improve the network's robustness [18], [19], [20].

In this paper, we propose *MultiSensory Dynamic Pretraining* (MSDP), a novel RL framework to learn expressive multisensory representations for contact-rich manipulation tasks. MSDP first learns to extract and fuse sensor features via an offline, pretraining phase based on masked autoencoding and cross-sensor prediction; then, an online RL agent leverages the features from the pretrained multisensory encoder through a new combination of cross-attention and pooling mechanisms applied to the critic and actor, respectively. In turn, this design enables RL policies to synergistically fuse multisensory inputs and inherently cope with sensor noise or missing modalities.

Our experimental results demonstrate that MSDP yields effective representations that accelerate RL on a variety of contact-rich manipulation tasks while being robust to sensor noise and changing object dynamics. The FT-sensor boosts MSDP's performance in two challenging real robot tasks by 14 % leading to near-optimal performance. Notably, the policy is trained directly in the real world on MSDP's multisensory latent representation, without any sim-to-real transfer. We achieve near-optimal performance in only 6,000 online interactions, which takes less than 55 minutes including data collection and pretraining. Beyond these results, MSDP scales naturally to an increasing number of diverse input modalities and only introduces a few learnable parameters in downstream training of RL agents.

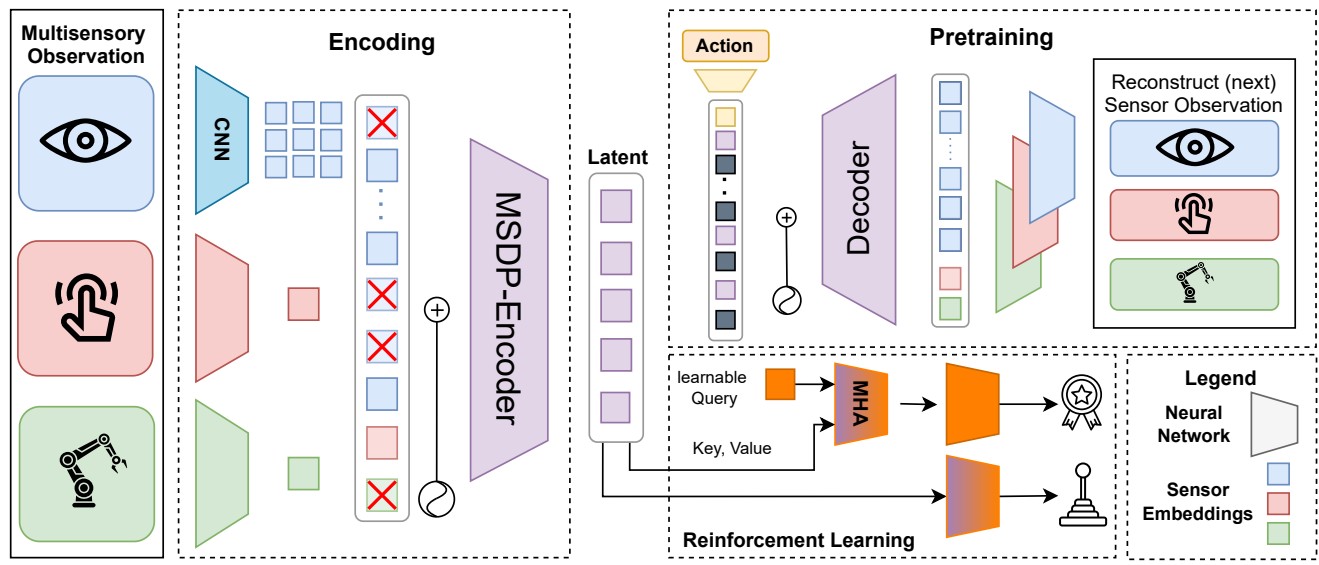

Fig. 2: The MSDP framework with MSDP-Encoder (left), Pretraining (top right) and downstream RL (bottom right): The current multisensory observation gets projected with a CNN-stem and linear layers to the embedding space. The MSDP-encoder fuses all sensor embeddings to form our expressive multisensory latent representation. The encoder is trained via (next) sensor observation reconstruction from a subset of sensor embeddings resulting in dynamic cross-sensor prediction. For downstream RL we extract multisensory task-specific features via a single cross-attention layer for the critic and via pooling for the actor. Our Framework offers an expressive and robust multisensory representation for complex contact-rich manipulation tasks in simulation and the real world.

To summarize, the key contributions of our work are twofold: (i) we develop an effective pretraining strategy based on masked autoencoding to form a rich multisensory representation, and (ii) we introduce a novel architecture that allows task-specific feature extraction to achieve efficient robot Reinforcement Learning. Our findings pave the way for adaptive and resilient RL agents to handle multiple input modalities to master complex contact-rich manipulation tasks.

## II. RELATED WORK

Tasks where a robotic manipulator has to interact with its environment, either directly or indirectly via a tool, require a good understanding of the interaction forces that shape the task at hand. Solving contact-rich tasks often relies on an accurate estimate of force and dynamics. When such estimates are available, classical control approaches can be adapted to solve the task [21], [22]. To better handle unseen or difficult contact dynamics in unstructured environments, RL presents itself as an ideal candidate that can learn directly via interactions [23], without accurate state estimation [24]. Various works have employed RL to learn contact-rich manipulation policies on a variety of tasks [25], [26], [27].

Lee et al. [4] learns a multimodal representation from vision, force torque, and proprioception via MLP-fusion and multiple self-supervised objectives. The frozen representation leads to a robust representation for RL to solve multiple peg insertion tasks. [14] extended the Vision Transformer [28] with a force torque sensor to solve a variety of contact-rich tasks. They additionally incorporate the self-supervised objectives from [4] to shape a representation using SLAC [29]. [30] pretrains an audio-encoder [31] to combine vision and audio via a transformer decoder for Imitation Learning. The

audio signal from the contact microphone provides rich feedback for various manipulation tasks. To also account for different sensor frequencies [13] developed a multi-resolution policy based on pretrained Vision Language Models to improve inference time using proprioception and force.

A robust representation is essential to integrate and process various sensory inputs to ensure stable and efficient learning in multisensory RL. Previous architectures often focused on straightforward latent fusion approaches by concatenating the various representations for downstream tasks [32], [33], [34], [35], [36]. Feng et al. [37] take this a step further by adding a subgoal-aware weighting for learning the stage-wise importance of difference sensors. Alternatively, contrastive learning approaches [38], [39] emphasize feature alignment, rather than fusion, to learn a shared latent representation between multiple modalities. However, distilling task-relevant features while maintaining sensor-specific information can be challenging. In contrast, inspired by the success of masked token prediction [16], recent works have also explored how masked multisensory pretraining can enhance representation learning for contact-rich manipulation [5], [15].

## III. MULTISENSORY DYNAMIC PRETRAINING

We present **Multi**Sensory **D**ynamic **P**retraining (MSDP), a novel framework for representation learning that builds upon Masked Autoencoders [40], tailored to enhance contact-rich robot reinforcement learning tasks that require perception through multiple sensor modalities. MSDP introduces a modular architecture to seamlessly handle diverse input sensors, and a multisensory masking scheme to promote rich cross-modal representations, i.e., to retain knowledge about the task and the environment dynamics even in the absence

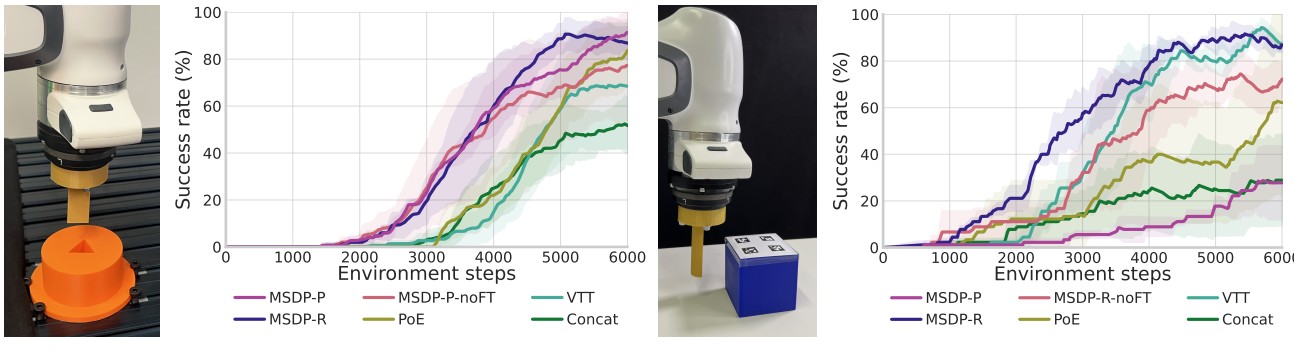

(a) Real World Peg Insertion.        (b) Real World Push Cube.

Fig. 3: Real world setup and experimental results. Our MSDP framework enables training RL policies directly in the real world, outperforming various baselines. Force torque readings are essential to consistently push the cube to the goal or to insert the peg under occlusion improving task success by 14 % (cf. MSDP-noFT). Policies are learned directly on the pretrained multisensory representation, without any sim-to-real transfer, with only 6,000 online interactions.

or perturbation of one or more modalities. Our framework decouples representation learning and downstream RL to ensure rich feature extraction and fusion, while offering stable, robust and compact downstream representations. Notably, our architecture is scalable w.r.t. the number and types of input modalities and also supports pretraining using additional sensors, which may not be available during policy training. As a result, our pretraining phase on limited offline data leads to rich representations that may be directly used to solve complex contact-rich RL tasks from high-dimensional input (including images) in less than $500k$ environment steps, whereas in the real world only $6k$ online samples are needed.

### A. MSDP Architecture

Given the inputs from multiple sensor modalities, such as vision, force torque readings and proprioception, we encode each modality using a separate network referred to as *sensor encoders*. Particularly for the vision input, we use a 4 layer CNN-stem similar to [41], which allows us to mask at embedding- rather than pixel-level, as masked object positions cannot be retrieved in pixel-space. The CNN-stem introduces redundancy as vision embeddings have overlapping receptive fields and, most importantly, stabilizes training [42]. We use a linear projection to encode force torque readings and proprioception to the 128-dim embedding space given their low dimensionalities. Subsequently, we add learnable parameters to each sensor embedding to contextualize the position and modality of it. Once we obtain embedding representations from all sensors, we randomly mask out a subset (70 %) of sensor tokens and feed the remaining to our 2 layer transformer encoder with 4 attention heads, pre-normalization and a mlp ratio of 2. The attention mechanism of the encoder leads to dynamic multisensory fusion promoted by the representation objective described in Section III-B. A learnable mask embedding is added to the randomly masked out embedding positions, next to the multisensory embeddings from the encoder, to generate the original number of embeddings. Separate decoder heads reconstruct the observation following the representation learning objective. We use a shared linear projection for all vision embeddings to reconstruct their corresponding patch.

### B. Representation Learning

Our representation learning objective is based on multi-modal masked autoencoding [40], [16], [17]. The objective is to either reconstruct the current $\mathbf{O}_t^{MS}$ or next observation $\mathbf{O}_{t+1}^{MS}$ from a random subset of sensor embeddings. Vision has a high number of embeddings and needs to extract information about other, potentially masked, sensors e.g. identifying contact to estimate force. Non-vision sensors on the other hand, are beneficial to reconstruct the vision observation as e.g. proprioception defines the robot position. Vision as a global sensing modality never gets fully masked out, whereas low-dimensional sensors are not available when masked. This representation objective results in cross-sensor prediction thus leading to fusion of all modalities.

### C. Policy Learning

From the pretrained encoder, we receive expressive multi-sensory embeddings given the available sensors. We name the mapping between embeddings and a compact representation *latent bridging*, which has a considerable impact on performance. To address this, we depart from works that naively extract the "CLS"-token from the high-dimensional embeddings [20], [28], [43], [44] and, instead, propose an asymmetric *latent bridging* strategy between actor and critic. The critic uses a single cross-attention layer with a learnable query and the multisensory embeddings from the MSDP encoder as keys and values. It offers dynamic task-specific feature extraction over the task-solving process. Fine-grained understanding of the environment leads to faster convergence compared to a global representation. The policy, on the other hand, does not profit from a cross-attention layer as it may destabilize training and receives the pooled representation from the sensor embeddings. Pooling sensor tokens results in a stable and parameter-free latent bridging similar to [5], [41]. This assymmetric representation between actor and critic, follows [45], where the actor benefits from a stable representation over task stages and the critic from a dynamic-specific representation. We train the policy with RLPD in the real world, where we incorporate offline data for pretraining in our replay buffer.

## IV. EXPERIMENTS AND RESULTS

In this section, we present our competitive baselines, real world setup and results.

### A. Baselines

We compare our methods against one transformer and two non transformer-based baselines. The latter models extract sensor-specific features with a CNN for vision and an MLP for proprioception and force torque readings. The **Concat** model fuses the concatenated features with a 2-layer MLP to form the multisensory latent representation [4]. The **PoE** model generates separate means and variances for each sensor and fuses them with a product of experts approach. **VTT** [14] fuses all sensors via a transformer-encoder and compresses the features via multiple linear layers to a compact latent.

### B. Real World Experiments

We conduct the Peg Insertion and Push Cube tasks in the real world using a Franka robot arm with a wrist-mounted FT-sensor and a custom endeffector. The observation space consists of endeffector position and velocity as proprioception, four force torque readings and an downsampled 64 by 64 RGB image from the 3rd person RealSense camera for both tasks. Peg and hole are both 3d printed. The robot needs to align the orientation of the triangular peg and place it precisely to fully insert the peg and complete the task. In Push Cube the agent needs to push the 7.5 cm block to the 15 cm away goal location. Successful episodes are detected with the endeffector position in Peg Insertion and an aruco marker and a second camera in the Push Cube task. The starting position of the robot is randomized and we use a sparse reward of +1 upon success in both tasks. An overview of the setup and results can be found in Figure 3a and 3b.

We build upon the SERL-package [46] and use a cartesian impedance controller for safe interaction. The multisensory encoder is pretrained with data from 20 demonstrations and ~2,000 samples of play data, resulting in a total of ~3,000 samples. Downstream tasks are trained directly on the real system for only 6,000 environment interactions with the proposed RLPD [47] algorithm from SERL. We pretrain for 6,000 steps in Peg Insertion, updating critic and actor twice per online sample. For Push Cube, due to higher vision variance, we use 10,000 update steps and apply four critic and one actor update step per online interaction to obtain meaningful representations and efficient training. The complete training pipeline, including data collection, pretraining, and online RL, learns the final policy in less than 55 minutes for both tasks. Compared to prior real-world multisensory RL methods [4], [14], our approach requires only a fraction of the data (3 % pretraining data in Peg Insertion) and training time (< 20 %) to learn the final policy.

Figures 3a and 3b show results for Peg Insertion (5 runs) and Push Cube (3 runs), respectively. Both our pretraining objectives achieve superior performance in Peg Insertion, whereas baselines suffer from vision noise and occlusion. Particularly, we notice that our synergistic use of force torque readings enables consistent insertion behavior, while reducing

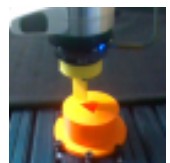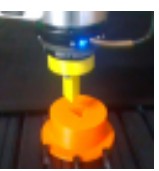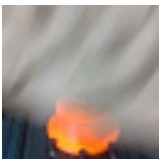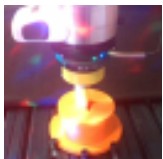

Fig. 4: Visual disturbances of the robustness evaluation. From left to right: back light, front light, occlusion, and disco lights. MSDP achieves a high success rate across all disturbances, which have not been seen during training.

the asserted force between the peg and the hole during exploration. On the other hand, the Push Cube task introduces several challenges as the agent needs to make and maintain contact with the cube without pushing it out of the workspace. Here, MSDP-R and VTT are able to obtain a high final success rate (see Figure 3b), while MSDP-P is not able to extract a suitable representation. We attribute this shortcoming to the complexity of learning forward dynamics in vision-demanding tasks on limited real world data and lack of observation histories. Overall, we highlight that our expressive multisensory representation, in combination with task-specific feature extraction via latent bridging, is able to obtain first success with less than 2,000 online interactions. FT-sensor usage particularly improves the performance of MSDP by 14 % in both real-world tasks, underscoring the significance of multi-sensor integration in contact-rich manipulation.

**Robustness Evaluation:** We evaluate the final policy of MSDP-P in the Peg Insertion task under various disturbances to showcase the robustness of our pretrained multisensory encoder and policy. We evaluate each change that hasn't been observed during training for 20 trials. Trained on $K_c = 2000$ cartesian-stiffness the policy achieves 90 % success rate with decreased ($K_c = 1500$) and 100 % with increased cartesian-stiffness ($K_c = 2500$). MSDP shows remarkable robustness against changed light settings, e.g., back light (100 %), front light (100 %), disco lights (100 %) and visual occlusion (partly blocked camera view, 95 %) and external forces (80 %). All visual disturbances are shown in Figure 4.

This work assumes the presence of all sensor modalities at the start of training to obtain an expressive representation from limited multisensory data. The usage of MSDP to enhance existing representations seems promising to leverage common available datasets for generalization.

## V. CONCLUSION

In this work, we propose MultiSensory Dynamic Pre-training (MSDP), a novel pretraining framework to learn multisensory representations for contact-rich manipulation tasks. Specifically, MSDP learns to reconstruct varying sensory information from a subset of input sensor embeddings, leading to cross-sensor predictions and sensor fusion. MSDP captures the interplay between different sensor modalities to learn a rich multisensory representation, which in combination with task-specific feature extraction through cross-attention, leads to a superior performance in challenging contact-rich manipulation tasks in the real world. The MSDP encoder and policy are robust against various disturbances e.g., controller parameters, heavy lighting variations, and external forces.

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
