# OpenReview forum: "Self-Supervised Multisensory Pretraining for Contact-Rich Robot Reinforcement Learning"
_IEEE.org/ICRA/2026/Workshop/Manipulation_Robustness — ICRA 2026_

### Official Review · Reviewer_oFNT · 2026-05-01
**Strong real-world multisensory RL with limited component attribution**

**Rating:** 7
**Confidence:** 4

**Review:**

This paper proposes MSDP, a masked multisensory pretraining framework for contact-rich robot reinforcement learning. Vision, force-torque, and proprioception are encoded as sensor tokens, pretrained through current or next observation reconstruction, and then used as frozen representations for online RL. The actor uses pooled embeddings, while the critic uses cross-attention over the pretrained embeddings. The method is evaluated on real-world peg insertion and cube pushing with a Franka robot.

Quality: The work is strong overall, especially because it validates the method on real contact-rich robot tasks rather than only on simulation or offline benchmarks. The reported gains from force-torque sensing are plausible, and the robustness tests under lighting changes, occlusion, stiffness changes, and external forces support the main claim that masked multisensory pretraining can improve resilience under common disturbances.

Clarity: The paper is mostly clear, but the source of the performance gain is not fully isolated. MSDP changes several components at once, including masked reconstruction, current-versus-next prediction, frozen embeddings, asymmetric actor-critic latent bridging, RLPD with offline data, and task-specific update schedules. More direct ablations of the latent bridging design and pretraining choice would make the argument cleaner.

Originality: The individual ingredients are related to prior masked pretraining, multisensory fusion, and real-robot RL, but the combination is meaningful in this setting. The most original part is the practical integration of masked multisensory representation learning with an asymmetric actor-critic architecture for contact-rich manipulation.

Significance: The significance is mainly practical. Achieving strong real-world performance with around 3,000 offline samples and 6,000 online interactions is valuable if the baselines are comparably tuned. The push-cube result is more mixed, since MSDP-P fails there, and the paper's claimed simulation evidence is not clearly visible in the provided experimental section.

Pros:

- Real-world contact-rich experiments with meaningful perturbation tests.
- Compact pipeline with learned multisensory fusion.
- Strong sample-efficiency claims on real hardware.
- Useful comparison against Concat, PoE, and VTT baselines.

Cons:

- Limited ablations for component-level attribution.
- Important hyperparameters are fixed without sensitivity analysis.
- MSDP-P fails on push cube, exposing a limitation of next-observation reconstruction.
- Simulation evidence is unclear despite being mentioned in the paper.

---

### Official Review · Reviewer_zQAQ · 2026-05-14
**Well motivated approach, limit**

**Rating:** 6
**Confidence:** 4

**Review:**

This paper proposes MultiSensory Dynamic Pretraining (MSDP), a masked-autoencoding framework for multisensory representation learning which is then used for training RL policy in the real world. The method pretrains a multisensory encoder by masking a subset of current sensor observations and reconstructing either the current or next multisensory observation, encouraging cross-sensor prediction across vision, force/torque, and proprioception. During downstream policy learning, the MSDP encoder is frozen and used to produce multisensory embeddings from the available sensor inputs. The actor receives a pooled representation of these embeddings to provide a stable, compact policy input, while the critic uses a single cross-attention layer over the frozen embeddings. The real-world experiments use RLPD and demonstrate strong sample efficiency, learning peg insertion and cube pushing in relatively few online interactions - beating several existing representation learning baselines.
Strengths
- The paper addresses a timely and important problem: current robot policies lack a standard mechanism for integrating heterogeneous modalities such as vision, force/torque, and proprioception.
- The masked-autoencoding objective is well motivated for multisensory encoder pretraining, since reconstructing missing sensor observations naturally encourages cross-modal prediction and sensor fusion.
- Real world results; demonstrating sample-efficient RL on contact-rich tasks is impressive and the simple approach is appealing to reproduce
Weaknesses
- The novelty of the pretraining objective is somewhat incremental relative to existing masked autoencoding and masked multimodal learning methods; the main contribution appears to be the real-world RL instantiation and actor/critic latent-bridging design. I would prefer to see more discussion compared to prior pretraining approaches that do masked reconstruction such as SPARSH (https://arxiv.org/abs/2410.24090), M2VTP (https://ieeexplore.ieee.org/document/10610933) or M3L (https://arxiv.org/pdf/2311.00924)
- The distinction between MSDP-R and MSDP-P is unclear. If MSDP-R and MSDP-P correspond to reconstruction and prediction, the paper should discuss why these objectives are separated rather than combined into a single objective.
- The paper would benefit from a stronger rationale for why cross-attention is used only in the critic and not the actor. The explanation that it may destabilize the actor is plausible, but stronger conditioning via CA makes more sense.

---

### Decision · Program_Chairs · 2026-05-21

Accept